# Is It Possible to Monitor Implant Stability on a Prosthetic Abutment? An In Vitro Resonance Frequency Analysis

**DOI:** 10.3390/ijerph17114073

**Published:** 2020-06-08

**Authors:** Paula López-Jarana, Carmen María Díaz-Castro, Artur Falcão, Blanca Ríos-Carrasco, Ana Fernandez-Palacín, José-Vicente Ríos-Santos, Mariano Herrero-Climent

**Affiliations:** 1Private Practice, Porto Dental Institute, Av. de Montevideu 810, 4150-518 Porto, Portugal; plopezjarana@gmail.com (P.L.-J.); arturfalcaoc@gmail.com (A.F.); 2Master’s Program of Periodontology and Implant Dentistry, University of Seville, Calle San Fernando, 4, 41004 Sevilla, Spain; carmmaria@hotmail.com (C.M.D.-C.); brios@us.es (B.R.-C.); 3Department of Social and Health Sciences, University of Seville, Calle San Fernando, 4, 41004 Sevilla, Spain; afp@us.es; 4Department of Periodontics and Implant Dentistry, Dental School University of Seville, C/Avicena S/N, 41009 Sevilla, Spain; 5Porto Dental Institute, Av. de Montevideu 810, 4150-518 Porto, Portugal; dr.herrero@herrerocliment.com

**Keywords:** resonance frequency analysis (RFA), implant stability quotient (ISQ), abutment, penguin RFA, insertion torque, implant stability

## Abstract

In order to apply the “one-abutment–one-time” concept, we evaluated the possibility of measuring resonance frequency analysis (RFA) on the abutment. This trial aimed to compare the Implant Stability Quotient (ISQ) values obtained by the Penguin^RFA^ when screwing the transducer onto the implant or onto abutments with different heights and angulations. Eighty implants (VEGA^®^, Klockner Implant System, SOADCO, Les Escaldes, Andorra) were inserted into fresh bovine ribs. The groups were composed of 20 implants, 12 mm in length, with two diameters (3.5 and 4 mm). Five different abutments for screwed retained restorations (Permanent^®^) were placed as follows: straight with 1, 2, and 3 mm heights, and angulated at 18° with 2 and 3 mm heights. The mean value of the ISQ measured directly on the implant was 75.72 ± 4.37. The mean value of the ISQ registered over straight abutments was 79.5 ± 8.50, 76.12 ± 6.63, and 71.42 ± 6.86 for 1, 2, and 3 mm height abutments. The mean ISQ over angled abutments of 2 and 3 mm heights were 68.74 ± 4.68 and 64.51 ± 4.53 respectively. The present study demonstrates that, when the ISQ is registered over the straight abutments of 2 and 3 mm heights, the values decrease, and values are lower for angled, 3 mm height abutments.

## 1. Introduction

Implant therapy has become the most important treatment for the replacement of missing teeth. One of the main factors to achieve osseointegration of dental implants is to have primary stability [1]. Primary stability is defined by the absence of mobility in the prepared bone site after implant insertion [2]. Moreover, the maintenance of an appropriate stability through time seems to be a long-term success guarantee [3,4,5]. The clinician can improve their decision tree in daily practice by assessing implant stability over time [6]. The possibility of stability monitorization and quantification gives objective information on implant behavior during osseointegration [7]. 

Different techniques have been described for the evaluation of implant stability. The best known, because they are non-invasive, are resonance frequency analysis (RFA) [8,9] and the insertion torque (IT) [10,11,12]. 

RFA assesses implant stability in an objective way at any stage of treatment or follow-up as it is highly reproducible and reliable [8,11,13,14]. This technique has been demonstrated to evaluate implant stability as a function of interface stiffness [9] and displays the micro mobility of implants (which seems to be determined by the bone density at the implant site, among other factors) [15]. Implant micromotions are important because, when they are above 150 μm, there is a high risk of fibro integration on the implant surface [16]. The unit of measurement in RFA is called the Implant Stability Quotient (ISQ), ranging from 1 to 100 (with 100 being the highest stability) [8]. 

It has been demonstrated that several factors influence the ISQ [17] values, such as the effective implant length, the distance from the transducer to the marginal bone (the further the transducer is to the bone, the lower the ISQ value), the osseous quality, the force with which the transducer is torqued, the presence of soft tissue between the implant and the transducer, and the amount of bone-in contact with the implant [4,11,18,19,20,21,22,23,24,25]. 

Another method to register primary stability, introduced by Johansson and Strid, is the insertion torque (IT) [26]. IT is the moment of force necessary to seat the implant into the osteotomy site. Its determination is done by a torque gauge incorporated within the drilling unit or with a torque wrench during the insertion of the implant [1,10]. It can only be measured once, when the implant is being placed, so the monitoring of the implant stability through IT is not possible [27]. IT measures the mechanical frictional resistance of the bone bed to apical implant advance, rotating about its longitudinal axis, whereas ISQ is based on the stiffness of implant contact with bone and therefore its resistance to lateral micromovements [6].

Because both methods evaluate the primary stability, researchers have studied what kind of correlation may exist between IT and ISQ [10,28]. The existence of a possible correlation between IT and ISQ values is quite a controversial topic. Several studies have shown a lack of correlation or minimal correlation between them at best [29,30]. However, these parameters measure two different mechanical concepts, the IT represent the axial resistance force of the bed bone preparation while the implant is inserted. The ISQ values represent the lateral stiffness between the implant surfaces and bone preparation [27,29,31]. However, the inverse relationship between ISQ values and micromotion has been previously documented [6]. Traditionally, the ISQ is measured by screwing the transducer directly onto the implant.

In the current age of increased importance on minimizing soft and hard tissue trauma, the “one-abutment–one-time” protocol was introduced as a minimally invasive prosthetic method [32]. In an experimental study, Abrahamsson et al. [33] showed that the repeated changes of prosthetic components (five times) caused an apical positioning of connective tissue and underlying bone. When the same research group limited the number of changes of the abutments to two times in another study, no differences in soft and hard tissues were observed between repeated and single abutment dis/reconnection [34]. With these findings in mind, the one-abutment-one-time concept was developed and implemented [32]. The concept was defined by Canullo as an immediate platform-switched restoration technique using only the definitive abutment [35,36,37]. Dedigi et al. in 2014 published a study where they compared a one abutment–one time test group with a control group where the abutment of the implant was removed at least three times (immediate implant with cemented restoration). An 87% increase of the mean recession of the buccal soft tissue in the control group was found [36]. With the knowledge that screwing and unscrewing the abutments several times could lead to a greater marginal bone loss [33,38,39,40], and in order to apply the one-abutment–one-time concept, transducers screwed onto the abutments were created, as opposed to previous transducers that have to be screwed directly onto the implant [41].

These new transducers are developed to avoid the dis/reconnection of abutments to assess the implant stability and to make ISQs registration easier (since it is not necessary to remove the abutment). As the transducer is placed more coronal (on top of the abutment), the assessment of the stability is simpler [33] and more convenient for the clinician. The last advantage of this method is that, in cases of low stability, there is no application of counterclockwise forces (implants with insufficient stability are not submitted to said forces) [42].

On the other hand, screwing the transducer onto the abutment (and not directly onto the implant) could affect the ISQ values, as the transducer is torqued farther from the bone than when it is screwed onto the implant. Moreover, screwing the transducer onto an angled abutment could significantly alter the ISQ values obtained. These modifications when conducting RFA could produce a greater vibration of the bone—implant interface and, therefore, a decrease in the ISQ values. For that reason, it is necessary to assess whether the ISQ measurements are comparable when using the transducers screwed directly onto the implant or onto the abutments [43].

The new system of RFA technology consists of a pen-like battery-driven device Penguin^RFA^ (Integration Diagnostics AB, Gothenburg, Sweden) and the transducer’s MulTiPeg™. These items are reusable (can be sterilized and used several times) and made out of titanium (the capacity to be tightened several times remains, unlike on items made out of aluminum). Penguin^RFA^ demonstrated excellent reproducibility and repeatability, so it could be suitable in attempts to monitor the stability of implants [24]. The aim of this trial was to compare the Implant Stability Quotient values obtained by screwing the transducer directly onto the implant or onto abutments with different heights and angulations. 

## 2. Materials and Methods

Type of study: This study was a descriptive transversal in vitro trial carried out at the Porto Dental Institute (Porto, Portugal). The sample size needed for this assay was determined by the Statistical Department of the University of Seville (Seville, Spain). The minimum sample size for ISQ values was statistically analyzed using N Query Advisor software (Statistical Solutions Ltd., Boston, MA, USA), setting the results from Herrero-Climent et al. [44] as a reference. For a statistical significance of *p* < 0.05 when comparing the two group means, with a two-tailed test assigned to the lower AFR means (66.87) and the upper one (73.74) with a standard deviation that included both 7140 and an estimated power of 80%, the required total sample size was determined to be of 18 (*n* = 18). To create four experimental groups that share their size, the total sample size was increased to 20.

Eighty implants (VEGA^®^, Klockner Implant System, SOADCO, Les Escaldes, Escaldes-Engordany, Andorra) with a rough surface obtained by subtraction and shot blasting were used. They were inserted into fresh bovine ribs simulating bone quality type II [45] and fresh femoral epiphysis simulating bone quality type III (by Leckholm and Zarb). The groups were composed of 20 implants with a 12 mm length and two diameters (3.5 and 4 mm). The implants consisted of an internal connection and double-threaded implants, characterized by an atraumatic apex and a progressive core. They were to be placed at a crestal level. 

Thus, 80 implants were placed and distributed as follows:-20 implants of a 3.5 mm diameter in type II bone.-20 implants of a 3.5 mm diameter in type III bone.-20 implants of a 4.0 mm diameter in type II bone.-20 implants of a 4.0 mm diameter in type III bone.

Five different abutments for screwed retained restorations named Permanent^®^, (Klockner Implant System, SOADCO) were then placed as follows: straight with 1, 2, and 3 mm heights and angulated abutments of 18° with 2 and 3 mm heights.

### 2.1. Surgical Protocol

The implants were placed by an experienced clinician following the manufacturer’s protocol. Ten implants (VEGA^®^, Klockner Implant System, SOADCO) were placed in each segment of bone. The osteotomy was performed under abundant irrigation with a sterile saline solution at 600 rpm. The implants were inserted using a calibrated dynamometer BTG90CN (Tohnichi, Tokyo, Japan), and placed at the crestal bone level. The distance between the implants had to be at least 10 mm in between the center of the implants. 

The Penguin^RFA^ (Integration Diagnostics AB) has different transducers called MulTipeg™ for different implant designs and connections (Figure 1). 

For VEGA^®^ with a 3.5 mm diameter, the MulTipeg Reference 57 was used; for VEGA^®^ with a 4 mm diameter, MulTipeg Reference 26 was used. Permanent^®^ abutments were screwed with metallic hand-screwdrivers with an IT of 5–10 N/cm^2^. The Multipeg^®^ used for the Permanent^®^ abutment was Reference 72 (Figure 2). The references of the Permanent^®^ abutments were 18-10-48 for the 1 mm straight abutment, 18-10-49 for the 2 mm straight abutment, 18-10-50 for the 3 mm straight abutment. 18-10-59 for the 18° angled 2 mm abutment, 18-10-60 for the 18° angled, 3 mm abutment. 18-10-62 for the, 30° angled 2 mm abutment, 18-10-63 for the 30° angled 3 mm abutment (Figure 3).

### 2.2. Variables

IT analysis was done when the implants were placed by means of a calibrated analogue dynamometer BTG90CN, so that an experienced clinician had the opportunity to register the exact IT. Once the implants were in place, primary stability was measured by means of RFA with two Penguin^RFA^ devices, by a peer of the first clinician. After the assessment of the RFA, the implants were removed (by counterclockwise forces), using a BTG90CN calibrated analogue dynamometer to register the disinsertion torque (DT). 

### 2.3. Protocols for Taking Measurements

ISQ values were obtained by taking two measurements with two different Penguin^RFA^ devices by an experienced clinician. These devices were proven to be reliable and allow for repeatable measurements of implant stability (Herrero-Climent [46]). In each situation, the ISQ value was registered perpendicular to the MultiPeg in two different positions: [1] the value was registered from the front of the bone segment at 90° to the vertical axis of the implant and always in the same direction and orientation of the device probe, and [2] the stability was registered from the right of the bone segment. At each position, the ISQ value was registered once with Penguin^FRA^ (Integration Diagnostics Sweden AB). The four ISQ values obtained were taken with the same transductor without its removal between measurements. An intraclass correlation coefficient (ICC) was analyzed to verify that the means could be used. Values A1-A2 are the first two repeated measures (Researcher 1), and B1-B2 are the second two repeated measures (Researcher 2). An ICC between both A values and between both B values was calculated to determine if they have any correlation so that we could use the stockings. In fact, since the ICCs were high, we used them (A) (B) and made an ICC again between the global A and the global B to see if we could use the global means of both researchers because they had a good correlation methodologically.

## 3. Statistical Analysis:

Mean values and standard deviations were calculated. The comparison of the numerical variables in the two groups was made with the Student’s *t*-test or the robust Welch’s *t*-test, when the data normality was verified or the Mann-Whitney test, in the absence of normality.

The comparison of the variables between more than two groups was performed with the analysis of variance (ANOVA) test or the ANOVA test with Welch correction (for normal data), or the Kruskal–Wallis test (non-normality). When significant differences were obtained, confidence intervals were found at 95%.

In each situation, four values of ISQ were registered. An ICC was analyzed to verify that the means could be used. As the ICC was very good (>0.90), a simple descriptive study of the means was carried out, including mean, standard deviation, and maximum and minimum values for data purification. This descriptive analysis was also carried out for IT and DI.

For inferential analysis according to diameter or bone, the normality of the distribution was investigated with the Shapiro–Wilk test, using a Student’s *t*-test for independent samples in the normal variables and the Mann-Whitney U test for the non-parametric ones.

For inferential analysis simultaneously taking into account both the diameter and the type of bone, an ANOVA test was performed involving the four groups, performing for those with a normal distribution the test of homogeneity of the variance of Levene with Anova according to Welch correction and subsequent multiple comparison tests. Those that do not meet normality were then looked into according to Kruskal–Wallis test, subsequently investigating the differences with multiple comparison tests.

## 4. Results

### 4.1. Resonance Frequency Analysis

The mean ISQ measured directly on the implant was 75.72 ± 4.37. The mean ISQ registered over straight abutments with a 1 mm height was 79.5 ± 8.50, that registered over straight abutments with a 2 mm height was 76.12 ± 6.63, and that registered over straight abutments with a 3 mm height was 71.42 ± 6.86. The mean ISQ over angled abutments of 2 and 3 mm heights were 68.74 ± 4.68 and 64.51 ± 4.53, respectively. Average ISQ values are according to height compared to those measured directly on the implant; (% increase or decrease in Table 1).

### 4.2. Insertion Torque/Disinsertion Torque

The insertion torque of the whole sample was 41.58 ± 11.70 N/cm. The disinsertion torque is 31.26 ± 12.14 N/cm. 

Regarding the repeatability of the Penguin ^RFA^, the mean intraclass correlation coefficient (ICC) was 0.917 for Multipeg A and B (Table 2). The ICC obtained between IT and DT was 0.576, and our results indicate that the higher the IT, the higher the DT. Bone quality showed an ICC of 0.442 and 0.638 for type II and III, respectively.

The first step involved analyzing the variables according to the diameter of the implants used (Figure 4). 

The IT values according to the diameter of the implants were 39.43 ± 12.98 Ncm and 43.73 ± 9.97 Ncm for implants with 3.5 and 4 mm diameters, respectively. The DT value for the implants with a 3.5 diameter were 35.55 ± 13.11 Ncm, and that for the implants with a 4 mm diameter was 27.79 ± 10.06 Ncm. There are statistically significant differences between IT values (*p* = 0.023) and DI values (*p* = 0.022) when comparing these variables according to the diameter of the implants. 

In the second step, the ISQ was analyzed grouping the implants according to the type of bone in which they were placed, as shown in Figure 5.

The IT values were 38.43 ± 9.87 Ncm and 44.73 ± 12.63 Ncm for bone types II and III, respectively. The DT values were 29.72 ± 8.00 Ncm for bone type II and 32.79 ± 15.16 Ncm for bone type III. There are statistically significant differences between IT values (*p* = 0.021) but not between DI values (*p* = 0.948) when comparing these variables according to the diameter of the implants. Figure 6 shows the ISQ values according to the combination of bone type and diameter.

In bone type II, the IT was 34.65 ± 9.06 Ncm for 3.5 mm diameter implants and 42.20 ± 9.38 Ncm for 4 mm diameter implants. In bone type III, the IT is 44.20 ± 14.62 Ncm for 3.5 mm diameter implants and 45.25 ± 10.54 N/cm for 4 mm diameter implants. There are statistically significant differences between IT values (*p* = 0.007). The DI values were 29.95 ± 8.13 Ncm and 29.47 ± 8.07 Ncm for 3.5 and 4 mm diameter implants, respectively, in bone type II. The DI values were 39.15 ± 15.56 Ncm and 26.11 ± 11.71 Ncm for 3.5 and 4 mm diameter implants, respectively, in bone type III. There are statistically significant differences between IT values (*p* = 0.022).

## 5. Discussion

In the current study, the implant stability was analyzed by RFA with the Penguin^RFA^ system when the transducer was screwed onto the implant and subject to different abutments of various heights and angulations. Penguin^RFA^ presents excellent reproducibility and repeatability [42,46]. In previous studies by Diaz-Castro et al. in 2019, Penguin ^RFA^ showed ICC values of 0.933 and 0.944 for Transducers 1 and 2, respectively [47]. The reproducibility was 0.906.

The systematic review of Atieh et al. in 2017 states that more frequent dis/reconnection in abutments would mean greater disruption to the peri-implant interface, and hence more bone loss should be expected. It also stated that disparities in the number of abutment removals, which ranged from one to four times, could be considered an important potential source of clinical heterogeneity [35]. Tallarico et al. included 14 articles in their metaanalyses and systematic review of randomized controlled trials. They found more biological complications in cases where a provisional abutment was used. They also reported significantly less mean bone loss (difference: 0.279 mm; *p* < 0.001) when the definitive abutment was used. The clinical implications are related to the pink aesthetic result of restoration, because when provisional abutments were used, greater buccal bone recession occurred, which is quite important in aesthetic areas (difference: 0.198 mm; *p* = 0.0004). Moreover, they recommended the routine placement of the definitive abutment with a simultaneous implant placement. This abutment should be carefully considered in cases in which more apically positioned restorative margins are needed. In this specific situation, the connection between the implant and the abutment, the implant position and macro-design, and an implant placement protocol seem to have the greatest influence on initial bone remodeling. Most of the scientific literature has confirmed that the use of a prosthetic procedure aimed at minimizing abutment disconnection and reconnection seems to decrease peri-implant bone-level changes [37].

Taking this philosophy into account, in 2017, Herrero-Climent et al. [48] compared the ISQ values obtained when measuring on the implant directly and when measuring over prototype healing abutments of 2, 3.5, and 5 mm heights. Similar values were found when measuring ISQ directly on the implant and on healing abutments of different heights. The ISQ mean values were 76.2 ± 4.47 when the ISQ was measured directly on the implant, 75.69 ± 4.7, and were 78.2 ± 5.78, 77.3 ± 5.9, and 76 ± 5.9 when the transducer was screwed to prototype healing abutments of 2, 3.5, and 5 mm. These results are because the healing abutments were designed to minimize the distance from the transducer to the bone [48]. Díaz-Castro et al. [47] found an ISQ value of 68.19 ± 3.31 using the same type of implants and with the same RFA system in a trial on fresh bovine rigs with a similar study design. 

The results of the current study showed that higher ISQ values are found when the measurements are done over straight abutments with a 1 mm height. These results could be explained by the fact that the abutments have a transmucosal height, so the oscillation force would be applied at a distance from the implant platform. A possible interpretation could be that, for the transducer to measure directly on the implant, it would need a longer lever arm from the top to the crestal bone (Figure 1). That aspect could influence the transmission signals to the bone in the RFA. It would be interesting to make a further trial comparing the same design for the transducer in order to study the corresponding values. Some articles have reported similar findings. Gültekin et al. [49] found that, when the transducer is further from the crest, the ISQ values are lower. A similar trend was observed in 2017 by Lages et al. [50] when analyzing the ISQ values of 31 external connection implants (4.1 × 10 mm) with the transducer screwed directly on the implant and on abutments of 1 and 5 mm heights. These authors reported ISQ values of 88.27 ± 5.70, 72.75 ± 4.73, and 66.67 ± 3.67, respectively (differences between groups were statistically significant), and concluded that the higher the transmucosal height increase, the lower the ISQ value obtained. 

Because of the differences in ISQ values depending on whether it is measured directly on the implant or on abutments of different heights and angulations, Lages et al. recommend always analyzing the ISQ values in the same way (either on platforms or abutments), so that the comparison of values over time is fairer. They also propose that studies should be developed to create a mathematical formula for finding the ISQ value measured on the platform and at different transmucosal abutment heights from just one of these values [51]. 

Taking into consideration the limitations of our study, we used the mean ISQ values in order to determine some mathematical relationship between the values depending on the abutment height. The direct implant mean ISQ values presented an increase of 4.9% over straight abutments with a 1 mm height. If we compare that with the mean ISQ value directly on the implant with a 2 mm height abutment, we obtain a 0.52% increase in mean ISQ values. However, we found for 3 mm a diminution of 5.67%, and for the angled abutment, a diminution of 9.21% and 14.80% for 2 and 3 mm, respectively (Table 2). In conclusion, as height increases, and above all, if used on the angled abutment, the mean ISQ values decrease (although with the advantage that the standard deviations are reduced, thus increasing the statistical homogeneity).

When the ISQ values were analyzed by the type of bone in which they are placed, the ISQ was lower for 3.5 mm diameter implants. These differences are significant except for when the RFA was done directly on the implant. These results are similar to the study carried out by Gültekin et al. [49], who found that ISQ measurements were significantly lower for implants with a 3.8 mm diameter (72 ISQ) than implants with a 4.6 mm diameter (79 ISQ). They found that larger diameters improve the bone implant contact and increase ISQ values. Nevertheless, Han et al., in 2010, compared a range of ISQ values for both implant diameters (+4.1 and +4.8 mm) and for both surfaces (SLA and SLActive) (Straumann^®^ AG, Basel, Switzerland), and found no significant difference among the three types of implants, meaning that the implant diameter is not a significant factor affecting ISQ values [25]. Our results suggest that the type of bone has no considerable influence over the ISQ values. It seems that ISQ values are similar between implants placed in bone type II and III (although slightly lower in type III), except when the registers are taken over angled abutments with a 3 mm height (ISQ values are statistically significantly lower for implants placed in bone type III) [52]. These results contrast from other findings, which indicated that higher ISQ values are observed for both implant systems when the cortical bone is maintained compared to when it is eliminated, the difference being statistically significant [53].

Therefore, the mean ISQ value measured directly on the implant was 75.72 ± 4.37. These results are similar to those published by Romanos et al. [54], in 2014, when they measured the ISQ of 3.3 × 10 mm bone level (Straumann) implants placed in cow ribs. They found a mean ISQ value of 75.02 ± 3.65. However, other trials have reported different results. In 2009, Andres-García et al. [55,56] found mean ISQ values of 70.86 ± 3.4 and 70 ± 3.8 when considering two different implants (3.7 × 10 mm Zimmer^®^ Dental and 4 × 10 mm NobelBiocare^®^) in cow ribs with bone quality types II and III. The difference between the results could be due to the different macro designs of the implants studied.

The possible correlation between the IT and ISQ values has been greatly discussed. However, these parameters measure two wildly different mechanical concepts. The IT represents the axial resistance force of the bed bone preparation while the implant is inserted. The ISQ values represent the lateral stiffness between the implant surfaces and bone preparation [27,29,31]. 

The results of this study showed that the ISQ value measured on angled abutments is lower than that registered directly on the implant and to that measured on straight abutments of different heights. It was also shown that an increase in height of the angled abutment results in lower ISQ values. 

The implications for clinical practice could be that the ISQ values can be determined directly on a Permanent abutment, at heights of 1, 2 or 3 mm, and that a variation of +/−5% can be obtained. This would avoid having to disassemble prosthetic abutments, which carries the risk of extracting the implant if the force of internal connection between the abutment and the implant is too high. Furthermore, when the ISQ values are determined directly on angled abutments, we must estimate that there is more distortion, which would be in a range between 10% and 15% depending on the height (2 or 3 mm on 18° abutments). A limitation of this study, in addition to the fact that it is an in vitro study, is the influence of the abutment design on the average values obtained, as they could vary from one trademark to another.

To our knowledge, this is the first trial that applies RFA over angled abutments, as we have not been able to locate similar studies in the literature that can be compared to our own. In view of our results, we deem it appropriate to state that further research should be done in a clinical practice setting in order to monitor the evolution of stability in RFA. All measurements should be taken on the definitive abutment so that they can be compared.

## 6. Conclusions

The present study demonstrates that when the ISQ is registered on straight abutments with 2 and 3 mm heights, the values decrease, and values are lower for angled, 3 mm height abutments.

## Figures and Tables

**Figure 1 ijerph-17-04073-f001:**
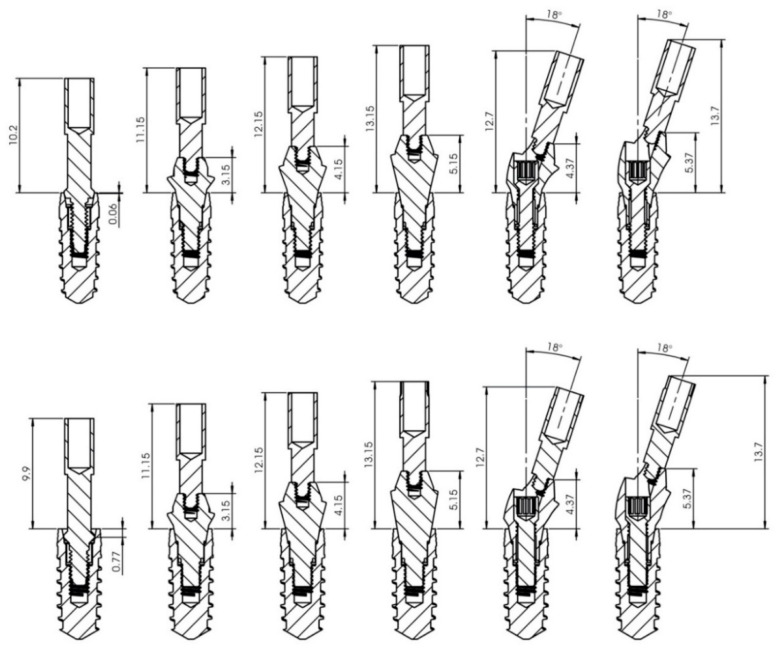
Abutment implant assembly, Permanent. Section 00, 3:1 scale. Implant Stability Quotient (ISQ) measurement sequence: directly on the implant; straight, 1, 2, and 3 mm height abutments; angled, 2 and 3 mm height abutments, respectively.

**Figure 2 ijerph-17-04073-f002:**
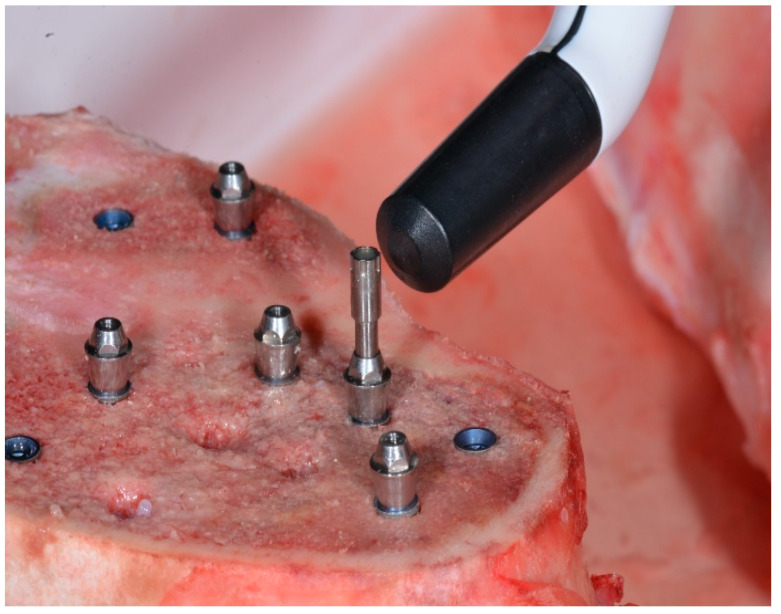
Resonance frequency analysis with Penguin RFA of Multipeg^®^ Reference 72 for the Permanent^®^, straight abutment with a 2 mm height for Vega^®^ with a 4 mm diameter).

**Figure 3 ijerph-17-04073-f003:**
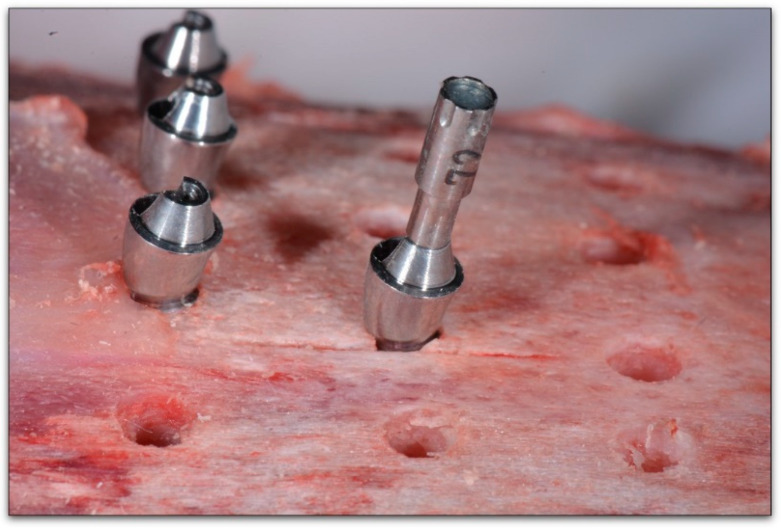
Resonance frequency analysis with Penguin RFA to Multipeg^®^ Reference 72 for the Permanent^®^, 18° angled abutment with a 2 mm height for Vega^®^ with a 4 mm diameter).

**Figure 4 ijerph-17-04073-f004:**
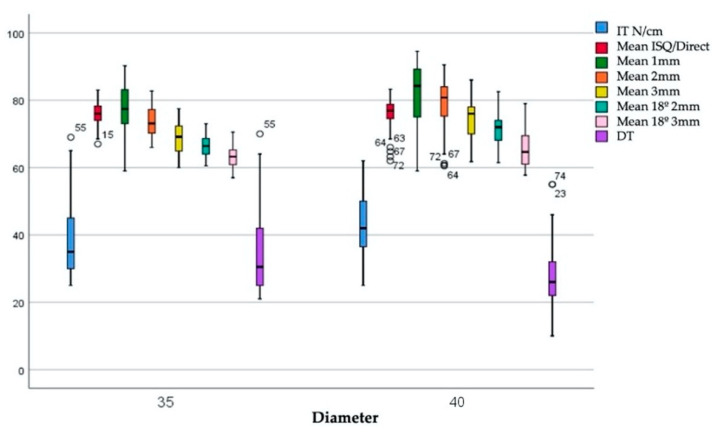
Insertion torque (IT) and disinsertion torque (DT). ISQ values with respect to the diameter.

**Figure 5 ijerph-17-04073-f005:**
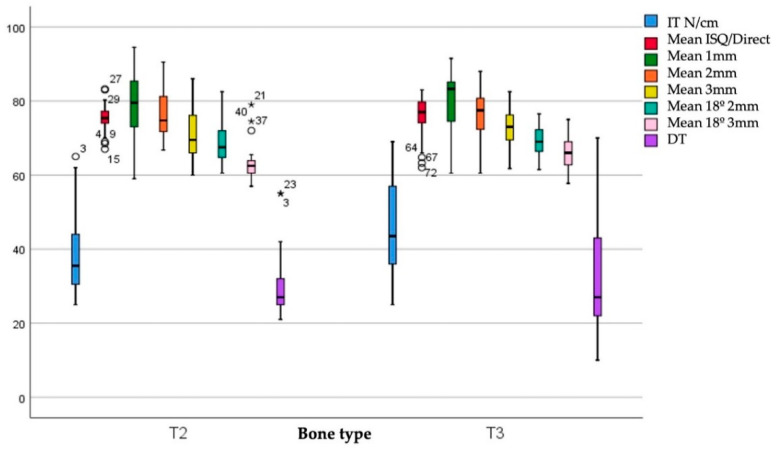
IT and DT. ISQ values according to bone type.

**Figure 6 ijerph-17-04073-f006:**
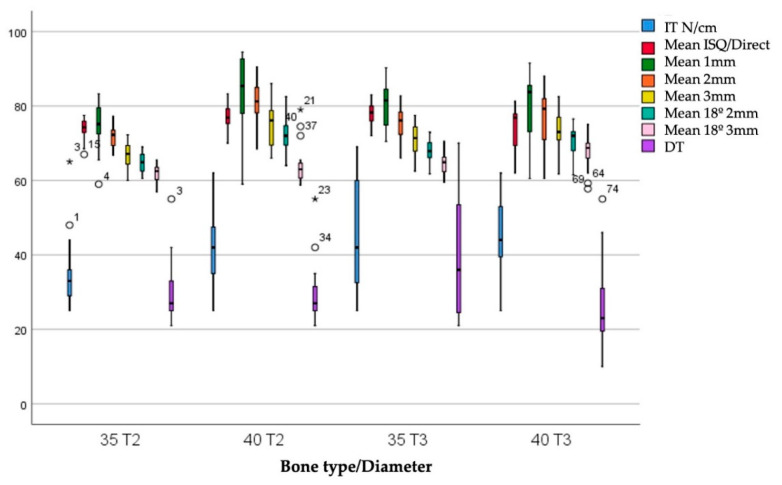
IT and DT. ISQ according to the combination of bone type and diameter.

**Table 1 ijerph-17-04073-t001:** Average Implant Stability Quotient values according to height compared to those measured directly on the implant (% increase or decrease).

Directly on the Implant	Straight, Permanent, 1 mm Height	Straight, Permanent, 2 mm Height	Straight, Permanent, 3 mm Height	Angled, Permanent, 2 mm Height	Angled, Permanent, 3 mm Height
75.72	79.5	76.12	71.42	68.74	64.51
	+4.9%	+0.52%	−5.67%	−9.21%	−14.80%

**Table 2 ijerph-17-04073-t002:** Intraclass correlation coefficient (ICC) for Penguin ^RFA^. Values A1 and A2 are the first two repeated measurements (Researcher 1), and B1 and B2 are the second two repeated measurements (Researcher 2). A mean ICC between both A values and between both B values was calculated.

	ICC A1-A2	ICC B1-B2	MEAN ICC A-B
**Directly on the implant**	0.93	0.84	0.96
**Straight, 1 mm abutment**	0.85	0.84	0.98
**Straight, 2 mm abutment**	0.87	0.87	0.99
**Straight, 3 mm abutment**	0.84	0.85	0.99
**Angled, 18°, 2 mm abutment**	0.74	0.57	0.95
**Angled, 18°, 3 mm abutment**	0.71	0.71	0.97

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
