# Peer review of "Is It Possible to Monitor Implant Stability on a Prosthetic Abutment? An In Vitro Resonance Frequency Analysis"

_ijerph, 2020, doi:10.3390/ijerph17114073_

Round 1
Reviewer 1 Report
The article presents interesting and up-to-date issue. However, there are many issues to be addressed. The manuscript demands English editing.
Title
I suggest changing the title into shorter one.
Introduction
Lines 93-104 should be summarized.
This section should be more focused on the topic of the research.
The name of the instrument should be removed from the aim. Please write the aim more clearly.
Materia and methods
Please add the approval of ethical committee.
Section “parameters”
Please rename it according methods used. And describe each method/parameter in details subsequently.
Please use one term for specific parameter measured though out the text.
Lines 173-177
Please remove points (1-3).
What was the influence of close implant placement on the results?
When was the measurement A1 and A2 performed?
The abbreviation ICC should be defined when used for the first time.
Results should not be presented in statistical analysis section.
Results
Results in tables 1-4 should be presented as one box and whisker plot.
The title of table 5 is not clear.
All IT and DT results should be presented as box and whisker plot.
Lines 201-202
What does the abbreviation CCI mean?
This section should be divided into paragraphs i.e. ISQ, ICC, DT, IT.
Discussion
This section should be more focused on the topic of the study.
Please add limitations of the present study.
Conclusions
This section should be rewritten. Please add clinical implications of this study.
Author Response
The article presents interesting and up-to-date issue. However, there are many issues to be addressed. The manuscript demands English editing.
The article had been extensively revised improving its English language.
Title
I suggest changing the title into shorter one.
Taking into attention your suggestions, title was changed to.
“Is it possible to monitor implant stability under the prosthetic abutment? An in vitro resonance frequency analysis.”
Introduction
Lines 93-104 should be summarized.
Taking into consideration your advices, these lines were eliminated.
This section should be more focused on the topic of the research.
We tried to reduce the extension of the introduction, and speak about the intentional objectives of the study.
The name of the instrument should be removed from the aim.
The name of the commercial instrument was eliminated.
Please write the aim more clearly.
The aim was reduced and clarified.
Material and methods
Please add the approval of ethical committee.
Our study was not subjected to the approval of an ethical committee. This in vitro study was developed on fresh rib bone bought on a butcher´s shop. Any animal was used or killed specifically for the purpose of the trial.
(All bone samples used were acquired commercially as products from animals destined for human consumption and therefore were not sacrificed specifically for this study)
Section “parameters”
Please rename it according methods used. And describe each method/parameter in details subsequently.
Please use one term for specific parameter measured though out the text.
Lines 173-177
Please remove points (1-3).
We have done it.
What was the influence of close implant placement on the results?
As with other similar studies, a minimum distance between implants placed of 10 mm was used in order to minimize the possible interference in ISQ measurements. Hypothetically the close proximity to another rigid structure as an implant could result in increased values of ISQ. Nonetheless as results compare values obtained over the same implant with different abutments, should there be any influence from implant proximity it doesn´t influence our results.
Georgios E Romanos , Gabriela Ciornei, Adina Jucan, Hans Malmstrom, Bhumija Gupta
In Vitro Assessment of Primary Stability of Straumann® Implant Designs. Clin Implant Dent Relat Res. 2014 Feb;16(1):89-95. doi: 10.1111/j.1708-8208.2012.00464.x. Epub 2012 Jun 12. PMID: 22691157 DOI: 10.1111/j.1708-8208.2012.00464.x
Herrero-Climent M, Santos-García R, Jaramillo-Santos R, Romero-Ruiz
MM, Fernández-Palacin A, Lázaro-Calvo P, Bullón P, Ríos-Santos JV.
Assessment of Osstell ISQ’s reliability for implant stability measurement:
A cross-sectional clinical study. Med Oral Patol Oral Cir Bucal. 2013 Nov
1;18 (6):e877-82.
Díaz-Castro MC, Falcao A, López-Jarana P, Falcao C, Ríos-Santos JV,
Fernández-Palacín A, Herrero-Climent M. Repeatability of the resonance
frequency analysis values in implants with a new technology. Med Oral
Patol Oral Cir Bucal. 2019 Sep 1;24 (5):e636-42.
When was the measurement A1 and A2 performed?
Values A1-A2 are the first two repeated measures (researcher1) and that B1-B2 of are the second two repeated measures (researcher 2). Line 189-190.
The abbreviation ICC should be defined when used for the first time.
The ICC was redefined.
Results should not be presented in statistical analysis section.
Statistical analysis section has references to the tables of results, but authors did not include the table inside the section. Tables is described on result section, and inserted at the end of the paper.
Results
Results in tables 1-4 should be presented as one box and whisker plot.
We have completed the statistical method and that we have changed it by following your instructions on the data tables using box charts and whisker plot. Personally, it seems to me loss of valuable data registered on the study; give us much more information before, but still visual.
Taking your considerations into account, there were included 3 whisker plot and 2 tables.
The whisker plots:
- Figure 4. Insertion Torque & Disinsertion Torque; ISQ values with respect of the diameter.
- Figure 5. Insertion Torque & Disinsertion Torque; ISQ values according to bone type.
- Figure 6. Insertion Torque & Disinsertion Torque; ISQ according to the combination of bone type and diameter.
The title of table 5 is not clear.
The table 5 express the Intraclass Correlation coefficient For Penguin RFA calculated on this paper.
- Values A1-A2 are the first two repeated measurements (researcher1)
- B1-B2 are the second two repeated measurements (researcher 2).
- A mean ICC between both A and between both B has been calculated.
All IT and DT results should be presented as box and whisker plot.
Lines 201-202
What does the abbreviation CCI mean?
CCI was a mistake, the right word is ICC (intraclass correlation coefficient).
This section should be divided into paragraphs i.e. ISQ, ICC, DT, IT.
Author divided this section into ISQ and DT/IT, taking into consideration your advice.
Discussion
This section should be more focused on the topic of the study.
The discussion has been summarized and centralized in the relevant aspects of the results obtained with each variable of the study. The objective of it is to present the results, compare them with the published evidence, as well as propose an explanation if the recruits differ.
Please add limitations of the present study.
A limitation of the study in addition to the fact that it is an in vitro study, is the influence of the abutment design on the average values obtained that could vary from one trademark to another ( line 356).
Conclusions
This section should be rewritten. Please add clinical implications of this study.
Conclusion statements were also reduced from 108 to 65 words
Clinical implications of the study were described on the final part of the discussion.
Reviewer 2 Report
Thanks for submitting this manuscript, which presents resonance frequency analysis taken directly to the prosthetic abutment.
I have carefully read your manuscript with great interest.
I think that it should sound very interesting for readers and this paper overall well written.
I found some typo-errors.
: ex)Line 225: 75,72 --> 75.22
Line 241: DI --> DT ??
Authors carefully need to check the typo-errors.
Sincerely,
Author Response
With thank you for your comments, authors have reviewed carefully typo-errors all over the paper.
Reviewer 3 Report
Please see my remarks

Author Response
Line 44-45
In this sense, implant stability monitorization can help deciding when to: initiate the rehabilitation stage, use an implant for an immediate loading, add an implant in a provisional restoration, etc.-
Is that a conclusion? please do not mention conclusion at that section.
This statement was eliminated.
Line 79
Do not mention such details at that section, please reduce that paragraph…
The paragraph was reduced from 407 to 234 words, as per your suggestion.
Line 93
Dedigi et al 2014, ref number missing.
We have introduced the reference number next to the author’s name, the reference was at the end of the paragraph.
Line 97 Do not mention such detail in that section.
The details were eliminated.
Line 126 Please remove that phrase.
That phrase was removed.
Line 156. Reference of the implant.
We have added the reference trademark of the implants, they are also described at line 139.
Line 167 Reference
Permanent® abutments were screwed with specific metallic hand-screwdrivers with an insertion torque of 5-10 N/ cm2.
18 10 48 ref permanent straight abutment (1mm)
18 10 49 ref permanent straight abutment ( 2mm)
18 10 50 ref permanent straight abutment ( 3mm)
18 10 59 ref RV 18º angled permanent abutment (2mm)
18 10 60 ref RV 18º angled permanent abutment (3mm)
18 10 62 ref RV 30º angled permanent abutment (2mm)
18 10 63 ref RV 30º angled permanent abutment (3mm)
Line 181 Who did estimate these measurements?
An experienced clinician.
Line 189 The values of p after performing of that model was?
the statistical method has been completely redone.
Linea 288 please replaced for showed that- done
Line 291 please do not repeat result: there were eliminated.
Line 295 How do we know that ref?: We have reduced that paragraph , in order to not to extent the discussion.
Line 309: authors: It was introduced the author related to that citation.
Line 350: reduce conclusion: the conclusions were reduced from 108 to 65 word.
Line 367: the clinical implications were inserted at the end of the discussion.
Round 2
Reviewer 1 Report
The manuscript has been improved. However, still English editing is mandatory.
Lines 150-152 need verification, as parentheses are in improper position.
Figures 4-6 should be moved to result section. The whole text in figures should be in English.
Do not use abbreviations is paragraph titles and in aim and in conclusion.
Conclusions need rewriting please present them as points (one or two). Now, conclusions resemble more results.
Author Response
With thank you for your comments, authors have reviewed carefully typo-errors all over the paper. We have changed the commas of the decimal places by points in all the wrong values. We have modified the abbreviations for the meaning in English.
Has been edited / revised in English through MDPI